# Association of Metabolomic Biomarkers with Sleeve Gastrectomy Weight Loss Outcomes

**DOI:** 10.3390/metabo13040506

**Published:** 2023-03-31

**Authors:** Wendy M. Miller, Kathryn M. Ziegler, Ali Yilmaz, Nazia Saiyed, Ilyas Ustun, Sumeyya Akyol, Jay Idler, Matthew D. Sims, Michael E. Maddens, Stewart F. Graham

**Affiliations:** 1Department of Nutrition and Preventive Medicine, Corewell Health William Beaumont University Hospital, 3601 W. 13 Mile Road, Royal Oak, MI 48073, USA; 2Oakland University William Beaumont School of Medicine, 586 Pioneer Dr, Rochester, MI 48309, USA; 3Beaumont Research Institute, 3811 W 13 Mile Rd, Royal Oak, MI 48073, USA; 4DePaul University Jarvis College of Computing and Digital Media, 243 S Wabash Ave, Chicago, IL 60604, USA; 5NX Prenatal Inc. Laboratory, 4800 Fournace Place, Suite BW28, Bellaire, TX 77401, USA; 6Allegheny Health Network, West Penn Hospital, 4815 Liberty Ave, Suite GR50, Pittsburgh, PA 15224, USA; 7Drexel University College of Medicine, 2900 W Queen Ln, Philadelphia, PA 19129, USA

**Keywords:** metabolites, sleeve gastrectomy, weight loss, obesity, bariatric surgery

## Abstract

This prospective observational study aimed to evaluate the association of metabolomic alterations with weight loss outcomes following sleeve gastrectomy (SG). We evaluated the metabolomic profile of serum and feces prior to SG and three months post-SG, along with weight loss outcomes in 45 adults with obesity. The percent total weight loss for the highest versus the lowest weight loss tertiles (T3 vs. T1) was 17.0 ± 1.3% and 11.1 ± 0.8%, *p* < 0.001. Serum metabolite alterations specific to T3 at three months included a decrease in methionine sulfoxide concentration as well as alterations to tryptophan and methionine metabolism (*p* < 0.03). Fecal metabolite changes specific to T3 included a decrease in taurine concentration and perturbations to arachidonic acid metabolism, and taurine and hypotaurine metabolism (*p* < 0.002). Preoperative metabolites were found to be highly predictive of weight loss outcomes in machine learning algorithms, with an average area under the curve of 94.6% for serum and 93.4% for feces. This comprehensive metabolomics analysis of weight loss outcome differences post-SG highlights specific metabolic alterations as well as machine learning algorithms predictive of weight loss. These findings could contribute to the development of novel therapeutic targets to enhance weight loss outcomes after SG.

## 1. Introduction

Obesity is a major global and national health challenge, with a US adult obesity prevalence of over 40% [1]. As a chronic disease, obesity is associated with multiple comorbidities, including type 2 diabetes, cardiovascular disease, stroke, and several types of cancer. The estimated annual medical cost of obesity in the United States is USD 173 billion. Annual medical costs for patients with obesity are ~USD 1861 higher than normal-weight patients [1]. Bariatric surgery is currently the most successful and durable treatment for morbid obesity, resulting in significantly greater weight loss compared to behavioral therapy and pharmacotherapy [2,3,4]. In addition to weight loss, there is a high percentage of obesity-related comorbidity resolution following bariatric surgery, including clinical resolution of type 2 diabetes, hypertension, and dyslipidemia [4].

Sleeve gastrectomy (SG) and Roux-en-Y gastric bypass (RYGB) are the two most commonly performed bariatric procedures. Both procedures demonstrate successful weight loss, as defined by at least 50% excess body weight loss, in most patients [5]. RYGB was historically the most predominant bariatric surgical procedure and preceded the SG procedure. The prevalence of SG has grown over the past decade, however, and has surpassed RYGB procedures. There has been a 451% increase in SG since 2011 in the United States, and SG comprised 61.4% of all bariatric procedures in 2018 [6]. Corewell Health East, formerly known as Beaumont Health, which is the health system the current study was conducted, reflects this trend with SG performed four times more frequently than RYGB over the past year. Despite the majority of patients achieving what is considered successful weight loss, there is variability in both the degree of weight loss and the propensity to regain weight. An understanding of the mechanisms and factors influencing this variability in weight loss may help predict bariatric surgery outcomes at an individual level, and potentially lead to targeted modalities to enhance weight loss outcomes.

Metabolomics is a field of study which involves the global investigation of metabolic pathways in biological systems with the focus on metabolites [7]. It involves high-throughput characterization and interpretation of the small molecules that are produced by cells, tissues, and microorganisms [8]. There is evidence to suggest that individuals with obesity have a metabolomic fingerprint that is distinct from normal-weight individuals [9]. Additionally, changes in metabolites occur after bariatric surgery. Two dominant metabolomic platforms include nuclear magnetic resonance (1H NMR) and mass spectrometry (MS). Due to the vast chemical and physical diversity of metabolites and the complexity of the metabolome, no single analytical platform can measure the concentrations of all metabolites. Hence, a combination of 1H NMR and targeted mass spectrometry coupled with liquid chromatography (LC–MS) can provide a more comprehensive understanding of the metabolome [10]. Additionally, large-scale quantification of metabolites, in a given biological system, enables researchers to identify an increased number of significantly perturbed biochemical pathways, their interconnectivity with other compounds (proteins, lipids, genes, etc.), environmental perturbations, and the microbiota [11].

A global metabolomics analysis is a comprehensive approach to characterizing metabolic changes associated with any given phenotype [12,13,14]. Metabolomic investigations to understand obesity and bariatric surgery outcomes are limited, and most focus on RYGB and analyze specific metabolites rather than employing a comprehensive metabolomics approach. There are few metabolomic biomarker studies that have evaluated metabolites in the context of weight loss outcomes. Kwon et al. evaluated whether baseline obesity-related amino acid metabolites were associated with weight loss in the early postoperative period in 27 patients after SG [15]. In this study, serotonin and the serotonin/5-hydroxytryptophan ratio showed superior performance in predicting slow weight loss six months after SG. These results suggest that preoperative amino acid metabolite profiles may be useful biomarkers for predicting early postoperative weight status after SG.

Metabolites have also been utilized in machine learning models to evaluate disease severity and predict outcomes to treatment [16,17,18]. Machine learning involves utilizing experimental data to develop and verify models that can be used for evaluating disease or predicting response to treatment. Statistical models are built on existing data to make reasonable predictions when presented with new data.

In our series of patients undergoing SG, we hypothesized that there would be an association between postoperative metabolomic profiles and weight loss outcomes. Additionally, we hypothesized that a machine learning algorithm could be developed using preoperative metabolomic data to predict weight loss outcomes post-SG.

## 2. Materials and Methods

### 2.1. Clinical Cohort

This prospective, single-arm observational study evaluated 45 patients with obesity entering one health system’s (Corewell Health, formerly known as Beaumont Health, Royal Oak, MI, USA) bariatric surgery program between February 2018 through April 2019 and planning to undergo laparoscopic SG. The cohort represents an interim analysis of an ongoing larger study of 150 patients with a 12-month postoperative follow up. The three-month post-SG follow up was completed in January 2020 and only patients that completed three-month follow up (N = 45) and submitted fecal and serum samples were included. The cohort was divided into tertiles, based on percent total body weight loss at three months post-SG, with 15 participants in each tertile. This study was approved by the Beaumont Institutional Review Board (IRB# 2017-201).

Inclusion criteria:Body mass index (BMI)○40 kg/m^2^ or above;○Or 35 to <40 kg/m^2^ with an obesity-related comorbidity, such as type 2 diabetes, heart disease, or obstructive sleep apnea [19].
Age 18–70 years old.Enrolled in the health system’s multidisciplinary bariatric surgery program.

Exclusion criteria:Inability to comply with regular post-operative follow-up visits,Pregnancy, andAny medical or psychiatric condition which in the opinion of the investigator makes the patient unlikely to be able to properly participate in this study.

### 2.2. Measures and Samples

Serum samples were collected before and three months after SG. Fecal samples were collected by participants at their residence via a stool collection kit before and three months after SG. The fecal samples were placed in a home freezer after collection for 24 to 48 h and then transported by the participant on ice, in an insulated bag, to research personnel. Preoperative sampling was performed before the patients started the calorie-restricted preoperative diet.

Serum and fecal specimens were stored in the biorepository at the Beaumont Research Institute at −80 °C. Demographic data were collected, weight and height were measured, and BMI was calculated at enrollment into the bariatric surgery program. Weight was measured again on the morning of SG surgery and at three months post-SG. Fecal and serum samples were collected preoperatively and at three months following SG.

Participants were divided into tertiles based on percent total weight loss (%TWL) at three months post-SG: (preoperative weight–post-operative weight at three months post-SG)/preoperative weight. Tertile 3 (T3) represents patients with the highest %TWL and tertile 1 (T1) represents those with the lowest %TWL.

### 2.3. Metabolomic Analyses

#### 2.3.1. ^1^H NMR Analysis Sample Preparation

##### Fecal Samples

Frozen fecal samples were thawed on ice, 65 (±5) mg transferred to microcentrifuge tubes and 600 µL of cold deionized water (4 °C) added. Samples were subsequently vortexed for 10 min, shaken at 200 rpm for 50 min at 4 °C. The mixtures were then centrifuged for 30 min at 15,000× *g* at 4 °C. Subsequently, 500 µL of supernatant was transferred to 3 KDa cut-off centrifugal filter units (Amicon Microcon YM-3; Sigma-Aldrich, St. Louis, MO, USA) and centrifuged at 13,000× *g* for 30 min at 4 °C. 200 µL of extract was combined with 25 µL of D2O and 21 µL of a standard buffer solution consisting of 11.7 mM DSS-d6 [disodium-2,2-dimethyl-2-silapentane-5-sulphonate], 1.75 M K2HPO4, and 5.84 mM 2-chloro pyrimidine-5-carboxylic acid (phasing standard) in H2O, containing 0.1% *w*/*v* sodium azide. The pH was measured and, if needed, adjusted to 7.4 and then centrifuged again. The final solutions were transferred to 3 mm NMR tubes for analysis.

##### Serum Samples

Serum samples were prepared using a modified version of the method as described by Mercier et al. [20]. A total of 300 µL of plasma were filtered through pre-washed (×7) 3.5 KDa filters (Amicon Micron YM-3; Sigma-Aldrich, St. Louis, MO, USA) via centrifugation at 13,000× *g*, at 4 °C for 30 min. To 228 µL of the filtrate 28 µL of D_2_O and 24 µL of 11.77 mM DSS-d6 in 50-mmol NaH_2_PO_4_ buffer (pH7) were added. Using a liquid handler system (Bruker Biospin, Billerica, MA USA), 200 µL of the mixture was transferred to a 3 mm NMR tube for analysis.

##### ^1^H NMR Data Acquisition

All ^1^H NMR experiments were recorded at 300.0 K (±0.05) using a Bruker Ascend III HD 600 MHz spectrometer coupled with a 5 mm TCI cryoprobe (Bruker-Biospin, Billerica, MA, USA). Using a randomized running order, 1D _1_H NMR spectra were acquired using a pulse sequence developed by Ravanbakhsh et al. [21]. More information can be found in the Appendix A.

#### 2.3.2. DI-LC/MS/MS Sample Preparation and Analysis

We employed the commercially available biocrates AbsoluteIDQ p180 kit (BIOCRATES, Life Science AG, Innsbruck, Austria) run on an Acquity UPLC I-Class (Waters, USA) coupled with a Xevo TQ-S (Waters, Milford, MA USA) to acquire our targeted and quantitative metabolomics data. More information is available in the Appendix A.

### 2.4. Statistical Analyses

All data were analyzed in Python [22]. A metabolite having more than 20% of missing values in any group was excluded from further analysis. In order to minimize sample to sample variation (dilution effect) and make individual features more comparable, raw metabolomics data were subjected to sum normalization, autoscaling and subsequently generalized log (g-log) transformed. Missing value estimation was carried out using the K-nearest neighbors (KNN) approach with K equal to 3. Principal component analysis (PCA) was performed to identify any potential outliers. A sample that deviated by more than 3 z-scores (i.e., 99.7% of confidence) of any of the first 3 principal components was considered an outlier.

#### 2.4.1. Univariate Analysis

Any metabolite which had >20% missing values in any group were excluded from further analysis. The remaining data were sum normalized and pairwise comparisons were calculated. These include T1: baseline (BL) vs. 3 months (3M), T2: BL vs. 3M, and T3: BL vs. 3M. Metabolite concentrations were analyzed using a Student’s T-test or a Wilcoxon signed rank test based on the results of their parametric test. For metabolites that were measured using both analytical platforms, an average value was used for all additional analyses. False discovery rates (FDR, q-values) were also calculated to account for multiple comparisons.

#### 2.4.2. Machine Learning Models

Several machine learning algorithms were evaluated to predict three-month post-SG weight loss outcomes, which included logistic regression, linear discriminant analysis, linear support-vector machine (SVM), and kernel SVM. Prior to examining the predictive performance of the algorithms and to ensure that there was no violation of the normality assumption sum, normalized metabolomics data were subjected to generalized log transformation (glog) and autoscaled. The data were split into training (60%) and testing (40%) sets. To select the best set of predictor variables utilized in the models and eliminate redundancy in the variable space, we employed the recursive feature elimination (RFE) [23] method with logistic regression as the classifier. Once the optimal set of metabolites were selected, some model hyper-parameters were optimized using 10-fold cross validation (CV). The details of the optimization and the function of the important parameters can be found in the literature [23]. The trained models were tested using 10-fold CV using a different seed. Accuracy, sensitivity, specificity, precision, F1-scores, and AUC metrics were calculated for each algorithm.

### 2.5. Metabolic Set-Enrichment Analysis

Metabolite set-enrichment analysis (MSEA) was performed using MetaboAnalyst (v5.0) [24] where metabolite names were converted to Human Metabolite Database (HMDB) identifiers. The raw data were subjected to quantile normalization, log transformation and autoscaled. The pathway-associated metabolite set was the chosen metabolite library, and all compounds in this library were used. Pathways with a *p*-value < 0.05 were significantly altered in both fecal and serum samples.

## 3. Results

Baseline characteristics of the patient cohort and the highest weight loss tertile (T3) versus the lowest weight loss tertile (T1) are shown in Table 1. The average %TWL at three months post-SG was 14.0 ± 2.6%. The %TWL for T3 versus T1 at three months post-SG was 17.0 ± 1.3% and 11.1 ± 0.8%, *p* < 0.001.

### 3.1. Serum Metabolomics

Univariate analysis within each individual tertile group at three months post-SG to baseline found no significant metabolite concentration changes. T3 metabolites at three months post-SG versus all patients at baseline, however, identified 49 metabolites that had a significant change in concentration (*p* < 0.05; Appendix A). Univariate analysis of T1 metabolites at three months post-SG versus all patients at baseline identified 70 metabolites that were at significantly different concentrations (*p* < 0.05; Appendix A). Both T3 and T1 had significant concentration increases in 3-beta-hydroxybutyric acid, acetone, and acetoacetate at three months post-SG relative to baseline. However, the degree of increase was greater for the T3 group than the T1 group, with increases of six- to twelve-fold versus three to five-fold, respectively. The T3 group had a four-fold decrease in methionine sulfoxide at three months post-SG whereas no change was observed in the T1 group.

Like the univariate analysis, comparison within each individual tertile group at three months post-SG to baseline found no significant serum metabolomic alterations. Comparison of each tertile to all patients at baseline, however, did reveal significant metabolomic alterations. There were 45 serum metabolites that were significantly perturbed for T3 at three months post-SG compared to all patients at baseline (*p* < 0.05; Figure 1; Appendix A). Comparison of T1 at three months post-SG with all patients at baseline identified 41 serum metabolites that were significantly perturbed (*p* < 0.05, Appendix A). Overlap was observed in the MSEA results when comparing both T3 and T1 at three months post-SG versus all patients at baseline. We found 36 metabolic pathways that were perturbed in both groups. Nine perturbed metabolic pathways were specific to T3, which included malate-aspartate shuttle (*p* = 7.62 × 10^6^), tryptophan metabolism (*p* = 3.23 × 10^5^), lysine degradation (*p* = 0.0001), glycine and serine metabolism (*p* = 0.0004), urea cycle (*p* = 0.001), aspartate metabolism (*p* = 0.003), methionine metabolism (*p* = 0.003), spermidine and spermine biosynthesis (*p* = 0.02), and betaine metabolism (*p* = 0.04). Conversely, five significantly perturbed biochemical pathways specific to T1 versus all patients at baseline were identified, which included sphingolipid metabolism (*p* = 0.001), pyruvate metabolism (*p* = 0.006), pyruvaldehyde degradation (*p* = 0.032), glycolysis (*p* = 0.042), and gluconeogenesis (*p* = 0.045).

Using RFE, we identified the top four predictive metabolites, which were acetone, hydroxybutyric acid, acetoacetate, and citric acid (Table 2), and evaluated 11 machine learning algorithms using their respective concentrations. The top four preoperative serum metabolites were highly diagnostic of three-month post-SG weight loss outcomes for the T3 group following a 10-fold cross validation (Figure 2). The top serum metabolite predictive algorithms included ridge classifier (AUC = 94.6%), Gaussian Naive Bayes (AUC = 93.5%), quadratic discriminant (AUC = 92.7%), and linear programming support vector machines (AUC = 91.6%). However, when we analyzed T1 versus all tertiles at baseline using the 11 machine learning algorithms, we were not able to produce any significant diagnostic models (Appendix A).

### 3.2. Fecal Metabolomics

Similar to the serum analyses, comparison within each individual tertile group at three months post-SG to baseline found no significant fecal metabolomic concentration changes or alterations. Univariate analysis of T3 at three months post-SG compared with all patients at baseline, however, identified a significant concentration change in 92 metabolites (*p* < 0.05; Appendix A). Univariate analysis of T1 at three months post-SG versus all patients at baseline identified a significant concentration change in 10 metabolites (*p* < 0.05; Appendix A). Significant fecal metabolite concentration changes that were specific to T3 included a five-fold decrease in both taurine and trans-4-hydroxyproline. For T1 relative to baseline, there was a significant three-fold increase in 3-hydroxyisovaleic acid and a two-fold increase in isoleucine, which were not observed in the T3 group.

Several metabolic pathways were found to be significantly perturbed for T3 at three months post-SG compared to all patients at baseline (Figure 1). Similar to the serum results, we observed an overlap in the MSEA results when comparing both T3 and T1 versus all participants at baseline, with both groups having alterations in bile acid synthesis. Three metabolic pathways were found to be specific to T3 versus baseline, which were arachidonic acid metabolism (*p* = 0.002), taurine and hypotaurine metabolism (*p* = 0.006), and fatty acid biosynthesis (*p* = 0.015). Conversely, two significantly perturbed biochemical pathways specific to T1 versus all patients at baseline were identified, which were porphyrin metabolism (*p* = 0.015) and methionine metabolism (*p* = 0.037).

Using RFE, we identified the top three metabolites, which were a phosphatidylcholine (PC aa C40:1), hydroxyoctadecenoylcarnitine (C18:1-OH), and a glycerophospholipid (PC aa C36:0) (Table 3), and evaluated 11 machine learning algorithms, as was previously done for serum. Following a 10-fold cross validation, we found the diagnostic accuracy was high when we compared T3 at three months post-SG with all tertiles at baseline. The top three predictive algorithms using fecal data included logistic regression (AUC = 93.4%), linear support vector machine (92.5%), and linear discriminant (91.4%; Figure 3). As observed in our serum models, when we analyzed T1 versus all patients at baseline using the 11 machine learning algorithms, we were not able to produce any significant diagnostic models (Appendix A).

## 4. Discussion

This comprehensive metabolomic analysis of weight loss outcomes in patients undergoing SG found significant changes in metabolite concentrations and metabolic alterations at three months post-SG, several of which were associated with the degree of successful weight loss. Additionally, we successfully developed machine learning algorithms, which demonstrated that preoperative serum and fecal metabolites were highly predictive of weight loss outcomes at three months post-SG. To our knowledge, this is the most comprehensive metabolomic study evaluating associations of metabolite changes with weight loss outcomes following SG. Furthermore, to our knowledge, this is the only study that has developed machine learning algorithms utilizing quantitative metabolomics data to predict weight loss outcomes.

Changes in serum and fecal metabolite concentrations at three months following SG that were specific to the highest weight loss group or had a greater degree of change relative to the lowest weight loss group included ketone bodies and amino acids. The highest weight loss group had greater increases in serum ketone bodies, including 3-beta-hydroxybutyric acid, acetone, and acetoacetate, relative to the lowest weight loss group. One explanation for these findings is that SG resulted in greater appetite suppression in the highest weight loss group, leading to greater calorie and carbohydrate restriction, which in turn led to ketosis and a greater increase in ketone bodies relative to the lowest weight loss group. We also found a five-fold decrease in fecal taurine concentration in the highest weight loss group at three months following SG. There is evidence that the amino acid taurine may be beneficial for weight loss through stimulation of energy expenditure, modulation of lipid metabolism, anorexic effect, and anti-inflammatory and anti-oxidative effects [25]. Taurine can be synthesized from methionine and cysteine. The decrease in taurine could potentially be related to the greater and more rapid weight loss in this group, which may have required higher utilization of taurine. In line with this explanation, there was a significant four-fold decrease in serum methionine sulfoxide as well as significant alterations in serum methionine metabolism in the highest weight loss group post-SG, which could be related to an increased need for taurine synthesis. Additionally, there were significant alterations in fecal taurine and hypotaurine in the highest weight loss group.

Other serum metabolite alterations specific to the highest weight loss group at three months post-SG included tryptophan metabolism, which is involved in regulation of inflammation. Fecal metabolite alterations specific to the highest weight loss group included arachidonic acid metabolism, which has been linked to the pathophysiology of obesity and the initiation and resolution of inflammation [26]. Serum metabolite alterations for the lowest weight loss group included sphingolipid metabolism and fecal alterations included porphyrin metabolism. Sphingolipids are implicated in a variety of inflammation-associated illnesses, including atherosclerosis, type II diabetes, and obesity [27]. Disorders of porphyrin metabolism have been associated with insulin resistance and metabolic syndrome [28].

Additionally, evaluation of machine learning algorithms found that both preoperative serum and fecal metabolites were highly predictive of weight loss outcomes, specifically for the highest weight loss group. The top three metabolite predictive algorithms accurately predict weight loss outcomes with an average AUC of 94.6% for serum and 93.4% for feces. The serum metabolites identified for the machine learning algorithms included acetone, hydroxybutyric acid, and acetoacetate, which are known to increase during ketosis. A fourth serum metabolite that was predictive, citric acid, is involved in the tricarboxylic acid cycle, which includes chemical reactions to release stored energy through the oxidation of acetyl CoA derived from carbohydrates, fats, and proteins into ATP. The fecal metabolites that were predictive of weight loss were the phosphatidycholine PC aa C40:1, hydroxyoctadecenoylcarnitine, and the glycerophospholipid PC aa C36:0. Phosphatidylcholines are the most abundant of the phospholipids and have several metabolic functions in organs, including the liver and the intestine. They have both structural and signaling functions in biological membranes and have been proposed to be involved in the metabolic effects of bariatric surgery [29]. Hydroxyoctadecenoylcarnitine is an acylcarnitine. Acylcarnitines transport organic acids and fatty acids from the cytoplasm into the mitochondria so that they can be broken down to produce energy. Glycerophospholipids have been shown to be altered post-bariatric surgery in a previous study and were associated with the lowering of risk factors for obesity-related diseases such as type 2 diabetes, nonalcoholic fatty liver disease and atherosclerosis [30].

Bile acids have been shown to be involved in regulating genes involved in obesity, lipid metabolism and inflammation. Clinical and animal studies have supported bile acids as potential key mediators of remission of chronic diseases that are associated with inflammation after bariatric surgery, such as type 2 diabetes [31]. Our study found significant alterations in bile acid biosynthesis at three months post-SG for both the highest and lowest weight loss groups. We did not assess, however, the change in concentration of bile acids for these groups.

Few studies have evaluated metabolomic changes in association with weight loss outcomes post-SG. Kwon and colleagues evaluated a limited number of metabolites, 20 amino acid metabolites, in 27 patients at three- and six-months post-SG [15]. They found that isoleucine and metabolites from the serotonin pathway were significantly associated with the percent excess weight loss at three and six months after SG. These metabolites were not found to be predictive of weight loss outcomes in our study. The metabolites that were identified in our machine learning algorithms, however, were predictive of the highest weight loss group, whereas the metabolites in Kwon’s study predicted the lowest weight loss group.

Limitations of this study include the short-term weight loss outcome of three months post-SG, rather than one year or greater. Early postoperative weight loss, however, has been shown to correlate well with long-term weight loss outcomes in previous studies [32]. Another limitation was the small sample size, which limited the power to detect metabolite concentration changes and metabolomic alterations within the same tertile group from baseline to three months post-SG. Therefore, we compared three-month post-operative metabolites to all patients at baseline. Our cohort was predominantly female and Caucasian, which limits the ability to generalize our findings across different groups. The T3 group had a lower percentage of women than the T1 group, at 73% versus 100%, which may have confounded the results. To investigate potential gender differences, a subgroup analysis was conducted exclusively on female participants in both the cohort and T3 groups for both serum and fecal data. The results of this analysis, which are presented in Appendix A and Appendix A, provide further support for the main conclusions of this study. Future studies could evaluate all female and all male cohorts to remove the impact of gender differences on weight loss outcomes. Additionally, some of the metabolite alterations identified, such as fecal bile acid biosynthesis, were not included in the analysis of metabolite concentration changes post-SG. Information on metabolite concentration changes that are associated with the metabolomic alterations identified would provide a more comprehensive understanding.

## 5. Conclusions

To our knowledge, our study is the most comprehensive metabolomic analysis of weight loss outcomes in patients undergoing SG. We found specific serum and fecal concentration changes as well as metabolomic alterations that were associated with disparate weight loss outcomes. We also identified machine learning algorithms that are predictive of greater weight loss outcomes at three months post-SG using both serum and fecal preoperative metabolites.

Further research on preoperative and post-SG metabolomics could lead to potential targets for perioperative intervention to enhance weight loss outcomes. Validation of machine learning algorithms in larger cohorts could enable more accurate predictions of anticipated weight loss post-SG, which will enable better informed treatment decisions. We plan to validate our findings with a larger cohort of patients and at time points further out from surgery.

## Figures and Tables

**Figure 1 metabolites-13-00506-f001:**
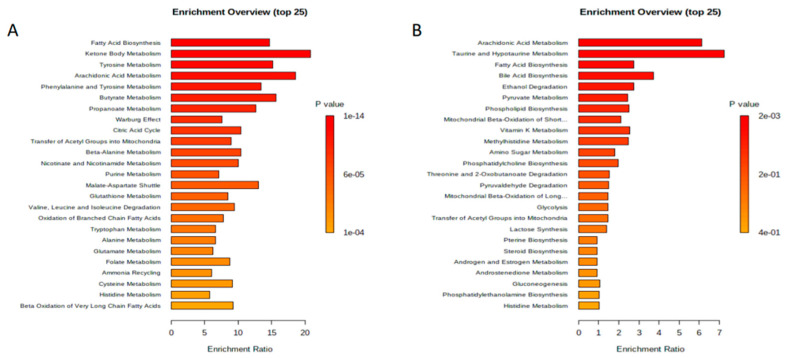
Metabolite set-enrichment analysis (MSEA) of the highest %TWL tertile (T3) at three months post-sleeve gastrectomy compared with all patients at baseline. (**A**) Serum. (**B**) Fecal. Note: A total of 209 fecal and 157 serum metabolites were used for the MSEA.

**Figure 2 metabolites-13-00506-f002:**
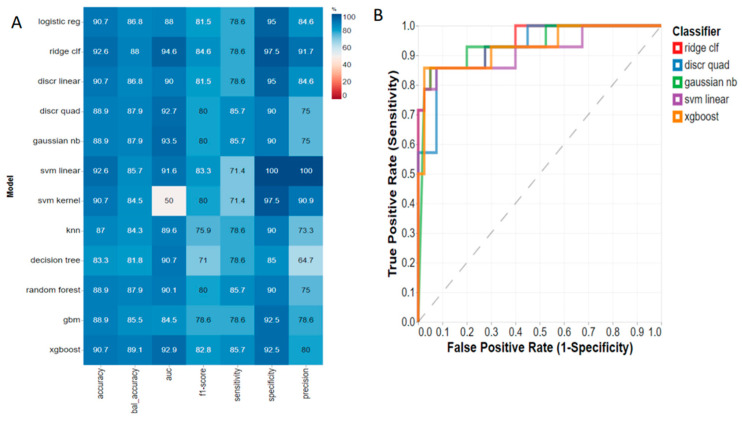
(**A**) Serum metabolite predictive models and (**B**) receiver-operating characteristic (ROC) plots for three-month post-sleeve gastrectomy weight loss outcomes: highest %TWL tertile (T3) versus all patients at baseline. Note: %TWL = percent total weight loss; logistic reg = logistic regression; ridge clf = ridge classifier; discr linear = linear discrimination; discr quad = discrete quad; gaussian nb = Gaussian Naive Bayes; svm linear = support vector machine linear; svm kernel = support vector machine kernel; knn = K-nearest neighbors; gbm = generalized boosted regression modeling; xgboost = extreme gradient boosting.

**Figure 3 metabolites-13-00506-f003:**
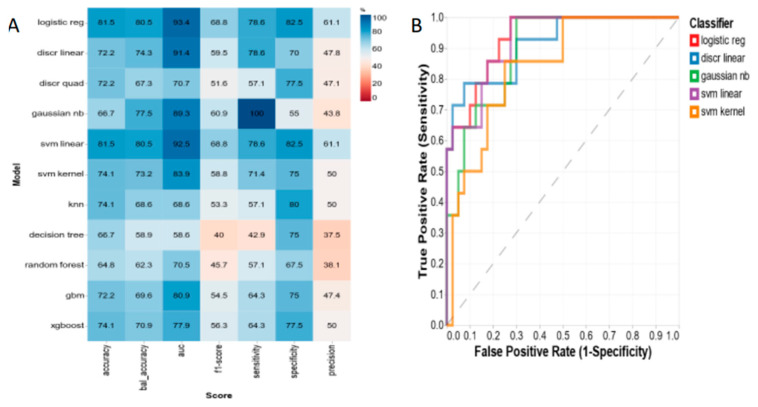
(**A**) Fecal metabolite predictive models and (**B**) receiver-operating characteristic (ROC) plots for three-month post-sleeve gastrectomy weight loss outcomes: highest %TWL tertile (T3) versus all patients at baseline. Note: %TWL = percent total weight loss; logistic reg = logistic regression; ridge clf = ridge classifier; discr linear = linear discrimination; discr quad = discrete quad; gaussian nb = Gaussian Naive Bayes; svm linear = support vector machine linear; svm kernel = support vector machine kernel; knn = K-nearest neighbors; gbm = generalized boosted regression modeling; xgboost = extreme gradient boosting.

**Table 1 metabolites-13-00506-t001:** Baseline demographics, anthropometric measures, and comorbidities, with comparison of the highest %TWL tertile (T3) with the lowest %TWL tertile (T1).

	Cohort (N = 45)	T3 (*n* = 15)	T1 (*n* = 15)	*p*-Value (T3 vs. T1)
Age (years)	48.2 ± 11.5	47.3 ± 12.1	51.0 ± 9.4	0.35
Weight (kg)	125.7 ± 20.6	123.8 ± 25.1	130.6 ± 18.7	0.38
BMI (kg/m^2^)	45.3 ± 7.3	42.0 ± 5.4	48.8 ± 8.7	0.02
Gender (female)	89%	73%	100%	0.10
Race				
- White	60%	73%	60%	0.43
- Black	36%	20%	40%	
- Two races	2%	7%	0%	
- Native American	2%	0%	0%	
Diabetes	29%	13%	40%	0.22
Dyslipidemia	56%	53%	47%	0.72
Hypertension	64%	53%	80%	0.12

Note: %TWL = percent total weight loss.

**Table 2 metabolites-13-00506-t002:** The top four preoperative serum metabolites selected for each of the machine learning analyses.

Tertile 1 (T1)	Tertile 2 (T2)	Tertile 3 (T3)
Hydroxybutyric acid	Hydroxybutyric acid	Acetone
Citric acid	Acetoacetate	Hydroxybutyric acid
Acetone	Citric acid	Acetoacetate
Acetoacetate	Acetone	Citric acid

Note: Tertile 1 = lowest % TWL; tertile 2 = middle %TWL; tertile 3 = highest %TWL.

**Table 3 metabolites-13-00506-t003:** The top three fecal metabolites selected for each of the machine learning analyses.

Tertile 1 (T1)	Tertile 2 (T2)	Tertile 3 (T3)
Phosphatidylcholine (PC aa C40:1)	Phosphatidylcholine (PC aa C40:1)	Phosphatidylcholine (PC aa C40:1)
Hydroxyoctadecenoylcarnitine (C18:1-OH)	Hydroxyoctadecenoylcarnitine (C18:1-OH)	Hydroxyoctadecenoylcarnitine (C18:1-OH)
Glycerophospholipid (PC aa C36:0)	Glycerophospholipid (PC aa C36:0)	Glycerophospholipid (PC aa C36:0)

Note: Tertile 1 = lowest %TWL; tertile 2 = middle %TWL; tertile 3 = highest %TWL.

## Data Availability

The metabolomics data represented in this exploratory study will be available upon request. Data is not publicly available due to privacy.

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
