# Peer review of "Association of Metabolomic Biomarkers with Sleeve Gastrectomy Weight Loss Outcomes"

_metabolites, 2023, doi:10.3390/metabo13040506_

Round 1

Reviewer 1 Report

I confirm that, this is the most comprehensive metabolomic study evaluating associations of metabolite changes with weight loss outcomes following SG. However,

  1. Authors should added the information about drugs, antibiotics, statins, supplements, MVI, etc.
  2. What with  nutritional questionnaire? No only BS, but diet and loss of weight before BS determines the type of surgery.
  3. What are the results when the cohort and T3 are all women? Because now is 89% female in cohort group and 73% in T3 group.
  4. Clinical significance of this study: validation of machine learning algorithms in larger cohorts could enable more accurate predictions of  anticipated weight loss post-SG, which will enable better informed treatment decisions. However, there is no discussion whether fast weight loss (T3) or slow weight loss (T1), better affects the level of metabolites that have different properties.

Minor comments:

  1. Authors should remove the format text page 2 line 91-100.
  2. Please check double spaces in all text.
  1. No all abbreviations are explained ex. DI-LC/MS/MS.
  1. Section 2.3.1. 1H NMR Analysis Sample Preparation – page 3 -  something wrong with specific sign.
  2. Please check the references, some positions need to be corrected.

Reviewer 2 Report

Dear Authors,

I was given a manuscript to check titled: Association of Metabolomic Biomarkers with Sleeve 2 Gastrectomy Weight Loss Outcomes

The thesis is written very meticulously, the methodology sufficiently explains the preparation of the study, the results are properly documented, and the discussion is sufficiently complemented by other studies... It is evident that the authors have a broad overview of the subject. I only have minor comments on the work.

-        I suggest providing the following modifications and additions:

-        Please delete lines 91-98. This is a recommendation for the authors.

-        Chapter 2.2. Measures and Samples (line 117) I recommend removing the fact that the "red top" tube color is irrelevant to the research presentation.

-        Make the inclusion criteria more transparent (for example, put them in bullet points) and add additional inclusion and exclusion criteria (lines 110-115).

-        Do not use an abbreviation in the titles of chapters 2.3.1 and 2.3.2,, which is not explained further in the text.

-        Also, define the abbreviations 1D and 1H NMR spectra (lines 159) and AUC (lines 203).

-        Include the number of probands in each group (Tertiles 1-3) and the number of probands who completed the follow-up after 3 months in Chapter 2.1 Clinical Cohort. 

Round 2

Reviewer 1 Report

Reviewer 1 Comments and Replies

Manuscript ID: metabolites-2231777

I confirm that, this is the most comprehensive metabolomic study evaluating associations of metabolite changes with weight loss outcomes following SG. However,

1.    Authors should added the information about drugs, antibiotics, statins, supplements, MVI, etc.

a)    Detailed analysis of the medications is beyond the scope of this project. Given the relatively small number of patients and the highly variable nature of their prescription and non-prescription medications, it was not felt we would be able to show anything with statistical validity.

b)   Exists many drugs and other chemical components which interfere with the lipid composition. Discussion about changes of lipids in serum without these data is useless.

2.    What with nutritional questionnaire? No only BS, but diet and loss of weight before BS determines the type of surgery.

a)    Please clarify this question.

b)   The authors should collect a nutritional questionnaire from patients, which informs about the diet and changes in eating habits. Do the authors have the mentioned nutritional questionnaires?

3.    What are the results when the cohort and T3 are all women? Because now is 89% female in cohort group and 73% in T3 group.

a)    The high percentage of females in our cohort was included as a limitation in the Discussion section:

b)     Lines 446 – 448: Our cohort was predominantly female and Caucasian, which limits the ability to generalize our findings across different groups. – I don’t see this sentence.

c)    We added the following to the Discussion section as a limitation:

d)    Lines 448 – 451: The T3 group had a lower percentage of women than the T1 group, at 73% versus 100%, which may have confounded the results. Future studies could evaluate all female and all male cohorts to remove the impact of gender differences on weight loss outcomes.

e)    The authors should show results when the study cohort is 100% women.

4.    Clinical significance of this study: validation of machine learning algorithms in larger cohorts could enable more accurate predictions of anticipated weight loss post-SG, which will enable better informed treatment decisions. However, there is no discussion whether fast weight loss (T3) or slow weight loss (T1), better affects the level of metabolites that have different properties.

a)    The scope of this project is to identify the association of metabolomic alterations and metabolite concentration changes with weight loss outcomes. It is unclear the degree that weight loss impacts the level of metabolites with different properties versus the degree that changes in metabolites impact weight loss outcomes. We provided evidence from previous literature on how metabolomic alterations that were found in our study are associated with weight loss outcomes and developed machine learning algorithms that could be tested in larger cohorts for validity. - I don’t agree. The authors determined which compounds change 3 m after SG. Based on their functions, it is possible to discuss which option is more beneficial for the patient.

b)    We did discuss T3 (greatest weight loss tertile) vs T1 (lowest weight loss tertile) and the types of metabolites that were affected in T3 relative to T1:

1. See Lines 374 – 393: Changes in serum and fecal metabolite concentrations at three months following SG that were specific to the highest weight loss group or had a

greater degree of change relative to the lowest weight loss group included…

2. See Lines 394 – 403: Other Serum metabolite alterations specific to the highest weight loss group at three months post-SG included…

Round 3

Reviewer 1 Report

I still believe that nutritional questionnaires and information about drugs, antibiotics, statins, supplements, MVI, etc. are necessary in metabolomics articles, especially in articles with patients with obesity and bariatric surgery, However,  I left the decision to the editor.

Author Response

"I approve the publication of the paper. However there is some formatting that need to be done, specillay throughout Section 2.3. Indeed in this Section the units are not appropriate in the version I read - volumes expressed in liter (l) instead of microliters (µl) I guess - lines 136, 138, 140, 141, etc

Response: Thank you for you suggestion. It has been updated in the Manuscript now. also, for the chemicals, lowercase and superscript should be used as appropriate - see lines 153, 154, 159, 161 etc H2O, D2O, 1D 1H NMR, K2HPO4, NaH2PO4"

Response: Thank you for you suggestion. It has been updated in the Manuscript now.